# New TEMPO–Appended 2,2′-Bipyridine-Based Eu(III), Tb(III), Gd(III) and Sm(III) Complexes: Synthesis, Photophysical Studies and Testing Photoluminescence-Based Bioimaging Abilities

**DOI:** 10.3390/molecules27238414

**Published:** 2022-12-01

**Authors:** Nataliya V. Slovesnova, Artem S. Minin, Anna V. Belousova, Aleksey A. Ustyugov, Kirill D. Chaprov, Alexey P. Krinochkin, Maria I. Valieva, Yaroslav K. Shtaitz, Ekaterina S. Starnovskaya, Igor L. Nikonov, Anton N. Tsmokalyuk, Grigory A. Kim, Sougata Santra, Dmitry S. Kopchuk, Emiliya V. Nosova, Grigory V. Zyryanov

**Affiliations:** 1Ural Federal University, 19 Mira Street, 620002 Yekaterinburg, Russia; 2Urals State Medical University, 3 Repina Street, 620028 Yekaterinburg, Russia; 3I. Ya. Postovskiy Institute of Organic Synthesis, UB of the RAS 22, S. Kovalevskoy Street, 620219 Yekaterinburg, Russia; 4M.N. Miheev Institute of Metal Physics, Ural Branch of the RAS, Russian Federation 18, S. Kovalevskoy Street, 620108 Yekaterinburg, Russia; 5Institute of Immunology and Physiology, Ural Branch of the RAS 106, Pervomaiskaya Street, 620049 Yekaterinburg, Russia; 6Institute of Physiologically Active Compounds at Federal Research Center of Problems of Chemical Physics and Medicinal Chemistry of the RAS 1, Severniy Proezd, 142432 Chernogolovka, Russia

**Keywords:** Eu(III) complex, TEMPO, 2,2′-bipyridines, EPR methods, pre-luminescence probe, biogenic thiols, BSA, fluorescence quenching

## Abstract

Linked to Alzheimer’s disease (AD), amyloids and *tau*-protein are known to contain a large number of cysteine (Cys) residues. In addition, certain levels of some common biogenic thiols (cysteine (Cys), homocysteine (Hcy), glutathione (GSH), etc.) in biological fluids are closely related to AD as well as other diseases. Therefore, probes with a selective interaction with the above-mentioned thiols can be used for the monitoring and visualizing changes of (bio)thiols in the biological fluids as well as in the brain of animal models of Alzheimer’s disease. In this study, new Eu(III), Tb(III), Gd(III) and Sm(III) complexes of 2,2′-bipyridine ligands containing TEMPO fragments as receptor units for (bio)thiols are reported. The presence of free radical fragments of the ligand in the complexes was proved by using the electronic paramagnetic resonance (EPR) method. Among all the complexes, the Eu(III) complex turned out to be the most promising one as luminescence- and spin-probe for the detection of biogenic thiols. The EPR and fluorescent titration methods showed the interaction of the resulting complex with free Cys and GSH in solution. To study the practical applicability of the probes for the monitoring of AD in-vivo, by using the above-mentioned Eu(III)-based probe, the staining of the brain of mice with amyloidosis and *Vero* cell cultures supplemented with the cysteine-enriched medium was studied as well as the fluorescence titration of *Bovine Serum Albumin*, BSA (as the model for the thiol moieties containing protein), was carried out. Based on the results of fluorescence titration, the formation of a non-covalent inclusion complex between the above-mentioned Eu(III) complex and BSA was suggested.

## 1. Introduction

Recently, among non-infectious “epidemics”, a number of diseases have been distinguished as the most threatening both economically and in their connection with the loss of living standards and working capacity of the people. These diseases are cardiovascular ones, cancer, chronic respiratory diseases, diabetes, neurological disorders, etc. [1]. For diagnostic testing of that pathology, the fluorescent dye could be used. In some cases, highly metabolic cells could be associated with sulfur-contained amino acids [2,3]. Changing in cysteine-contained peptide was found in tissues and organs in different pathology: from alcoholic liver disease [4] to neurological conditions, such as Huntington’s disease [5]. Cysteine metabolic pathways are discussed in connection with different cancer variants (for example, adenocarcinoma [6,7], Ewing Sarcoma [8], etc.). Cysteine and glutathione visualization can be used for cell metabolism in various non-cancer diseases. The biogenic molecules associated with ROS decrease the system during diabetic neuropathy [9], obesity [10], and cardiometabolic diseases [11].

Another disease that can be identified by means of monitoring the cysteine levels is Alzheimer’s disease. That disease is considered the most common cause of dementia [12].

The development of AD is associated with an increase in the deposition of various abnormal peptides: *beta*-amyloid (Aβ, amyloid theory) and *tau* protein (an overly phosphorylated protein involved in the movement of microtubules-tau-peptide [13]). *Beta*-amyloid (type β42) is contained in the intercellular space and cerebrospinal fluid of healthy people in the amount of 10% of all *beta*-amyloids. In people with Alzheimer’s disease, this protein forms aggregates in the intercellular space-amyloid plaques. Abnormal *tau* protein forms mainly intracellular inclusions-neuro-fibrillar tangles [14]. Amyloid proteins contain cysteine residues, namely 2–4 units per 220 proteins for plaque amyloid [15], 2 units per 108 residues in the amyloidogenic dimer [16], and 18 units for amyloid-beta precursor protein [17,18].

Large accumulations of amyloid and/or *tau* protein can be visually detected, while at the early stages of AD (before significant degeneration of the nervous tissue), the diagnosis is difficult [19]. And the detection of peptide clusters as a separate task can be achieved by using biomarkers [20], by using radio-chemical-pharmaceutical approaches [21,22], by using nanoparticles [23], or, finally, by using fluorescent dyes or probes.

Based on the all mentioned above one, interactions between biogenic thiols, including cysteine, and fluorescent dyes may be used for the early diagnosis of various pathologies, including Alzheimer’s disease.

Depending on the nature of the “(bio)thiol(for instance, cysteine)-dye” interaction, the mechanism of its detection can be different. For example, detection via additional reaction of thiols to alkene moiety in NIR xanthene-benzothiozolium dye [24], “AIE + ESIPT” probes [25], (bio)thiols ratiometric fluorescent detection via π-conjugation modulation based on spirocyclic open-closing molecular switch [26] as well as others [27]. Another sensing approach for cysteine-containing (macro)molecules detection is based on using so-called “pre-fluorescent” probes bearing nitroxyl radicals in their structure with fluorescence restoration when these radical probes are converted to a hydroxylamine derivative, for instance, upon hydrogen abstraction [28,29] or upon recombination with free radicals [30,31]. On the other hand, several lines of nitroxyl radical-appended lanthanide (III) complexes of 2-(3-pyridine-6-methoxyl)-4,4,5,5-tetramethylimidazoline-1-oxyl-3-oxide have been reported as a sensitive sensor for the iron (III) [32,33], as magnetic substances [26], luminescent single-molecule magnets (SMMs) [34], etc. A number of TEMPO-containing lanthanide(III) complexes have been reported, such as Er(III)-based molecular magnets [35], Eu(III)-based framework as a catalyst for the aerobic oxidation of alcohols [36], etc.

In this article, we wish to report our approaches for the detection of the above-mentioned biothiols by using TEMPO-appended 2,2′-bipyridine-based Eu(III), Tb(III), Gd(III) and Sm(III) complexes.

## 2. Results and Discussion

### 2.1. The Synthetic Design of the TEMPO-Lanthanide(III)-Based (bio)Thiol Probes

The main idea of this work is based on the synthetic design of new lanthanide(III) complexes with organic ligands bearing TEMPO moieties due to the ability of the last ones to react selectively with thiol groups [37]. Widely known 2,2′-bipyridines were selected as ligands due to their ability to form stable complexes with lanthanide (III) cations in the presence of additional rigid chelating groups [38]. In this case, the TEMPO moieties appended to the 2,2′-bipyridine would remain free after the lanthanide (III) complex formation and would cause the “pre-luminescence”/low-luminescence state of the whole complex due to the influence of free radical moieties. In the presence of (bio)thiols, the “turn-on” luminescence response of the pre-luminescent lanthanide (III) complex would be expected due to the “nitroxyl radical-thiol” interactions.

### 2.2. Synthesis of New TEMPO-Appended 2,2′-Bipyridine-Based Eu(III), Tb(III), Gd(III) and Sm(III) Complexes

Derivatives of 5-aryl-2,2′-bipyridine-6-carboxylic acids were used as ligands for lanthanide cations. Previously, lanthanide (III) complexes based on ligands of this series with a composition of 3:1 were prepared, and they exhibited up to 11% quantum yields of the luminescence of the europium(III) cation [38], and the synthesis of these ligands was performed by using a so-called “1,2,4-triazine” methodology [39,40]. Namely, the cyano group, which was introduced via direct C−H-functionalization [41,42] in the series of 1,2,4-triazine-4-oxides, acted as a synthetic precursor of the carboxyl group. Subsequent *aza*-Diels--Alder reactions with such dienophiles as 2,5-norbornadiene or 1-morpholinocyclopentene with the following hydrolysis of the cyano group led to the formation of the target ligands.

In the frame of this work, the 2-pyridyl fragment of the considered system was functionalized to introduce the TEMPO residue. Namely, an ester group was introduced at the initial stage of synthesis, opening up the possibility of its further functionalization. Previously, we reported the possibility of obtaining various 5-aryl-3-(2-pyridyl)-1,2,4-triazine-4-oxides functionalized at the 2-pyridyl residue, including those containing 5-methoxycarbonylpyridin-2-yl moiety in the position of C3 [43]. We have also shown the possibility of direct cyanation of these compounds. In accordance with the previously reported procedures, we synthesized 1,2,4-triazine-4-oxide **1** by using the reaction between *iso*-nitrosoacetophenone hydrazone **2** [43] and 5-methoxycarbonylpyridine-2-carbaldehyde **3**. Further cyanation via direct C-H functionalization reaction led to the 1,2,4-triazine-5-carbonitrile **4**. The subsequent *aza*-Diels--Alder reaction using 1-morpholinocyclopentene as a dienophile [44] resulted in a new cyano-substituted of 2,2′-bipyridine **5**. Further transformation of the ester group via its reduction with the following oxidation with MnO_2_ of thus obtained alcohol **6** led to the 2,2′-bipyridine-5-carbaldehyde **7**. These modifications were performed in accordance with our previously developed procedures for methyl 5-phenyl-2,2′-bipyridine-5′-carboxylic acid ester [45,46].

The subsequent reaction between the aldehyde **7** and 4-amino-TEMPO made it possible to obtain Schiff’s base **8**. In order to preserve the nitroxyl radical in the TEMPO composition, the further hydrolysis of the cyano group was carried out under mild conditions, such as boiling in an aqueous-alcoholic medium in the presence of Cu^2+^ salts as previously reported [30]. As a result, the Cu(II) complex **9** was obtained. The target lanthanide complexes **10** were obtained by using the previously described approach [38] via the treatment of copper(II) complex with in situ generated cyanide anions and the reaction of the resulting potassium salt with the corresponding Ln(III) chlorides to afford the target complexes **10** (Figure 1).

### 2.3. Determination of the Structure of the Obtained Semi-Products and Lanthanide (III) Complexes

The structure of compounds **5**–**7** was confirmed on the basis of ^1^H and ^13^C NMR spectroscopy (Appendix A), mass spectrometry and elemental analysis. In particular, in the case of aldehyde **7**, in the ^1^H NMR spectrum, a characteristic signal of the proton of the aldehyde group in the region of 10.2 ppm was observed, signals of the protons of the cyclopentene fragment in the region of the resonance of aliphatic protons, and, finally, the signals of protons of the ABX system of the pyridine ring. The presence of TEMPO moieties in the structure of the Schiff base **8**, copper complex **9**, and target lanthanide complexes **10** was confirmed by means of elemental analysis, mass spectrometry and EPR spectroscopy (Figure 1 and Appendix A). In particular, in the mass spectra (ESI-MS) of all the complexes **10**, the presence of peaks of the corresponding molecular cation [M + H]^+^ is detected. The isotopic distribution in all cases corresponds to the expected one, which also confirms the formation of the target complexes containing nitroxyl radical moieties. The recorded EPR spectrum of complex **10a** in DMSO corresponds to the EPR spectrum of the nitroxyl radical with a reduced rate of isotropic rotational diffusion, possibly due to the size of the molecule and hyperfine interaction constants (HFI) equal to 1.578 mT and 1.541 mT with broad unresolvable lines 0.252 mT wide [47].

The EPR spectrum of compound **10a** in THF corresponds to the nitroxyl radical with HFI constants of 1.555 mT and 1.545 mT, with a linewidth of 0.29 mT. The linewidth is due to unresolvable spectral lines indicating additional HFI constants with values close to the spectral linewidth. The spectral line shapes indicate a higher rate of isotropic rotational diffusion than THF.

The EPR spectrum of complex **10a** in D_2_O corresponds to nitroxyl radical with HFI constants of 1.690 and 1.648 mT with a reduced rate of isotropic rotational diffusion and broad (0.21 mT) unresolvable spectral lines.

The simulated spectrum of the nitroxyl radical [47] looks similarly based on the EPR spectroscopy data for complexes **10b**–**d** (Figure 1). Based on the simulation results, it was shown that the correlation time of the rotation of molecules in aqueous solutions is 10^−10^ s. Figure 1 on the left shows the simulated spectrum of the nitroxyl radical corresponding to the EPR spectrum of the Eu(III) complex **10a**. Thus, the difference in the width of the first and third spectral lines indicates the correspondence to the simulated spectrum and the rotation correlation time equal to 10^−9^ s. The difference in the correlation time between the Eu (III) complex **10a** and the rest of the lanthanide (III) complexes **10b**–**d** may be due to the different positions of the ligands in the complex relative to the coordination center.

In addition, compound **9** and the potassium salt of TEMPO-containing 5-phenyl-2,2′-bipyridine-6-carboxylic acid obtained in the course of in situ synthesis also exhibited a well-pronounced EPR activity. Moreover, in the first case, one can also observe the signal of the copper (II) cation in the range of 3000–3450 gauss [48,49,50], which confirms its presence in the composition of complex **9**. In this compound, the copper cation is less susceptible to the loss of the magnetic moment compared to the europium (III) cation in complex **10a** due to a significantly smaller number of bonds with the environment, namely ligand and chloride anion (Figure 2).

Next, by using the Eu(III) complex **10a** as the most representative example, we studied the EPR activity of nitroxyl radicals in various solvents over time. In particular, in the DMSO medium, the partial degradation of nitroxyl radicals was observed in 80 min, which is expressed in a proportional decrease in the intensity of the corresponding signals (Figure 3). The partial disappearance of the signals of nitroxyl radicals, apparently, is explained by the tendency of DMSO to interact with these radicals with the subsequent formation of carbon-centered methyl radicals, which recombine nitroxyl-based radical moieties irreversibly [51,52]. Thus, based on the data obtained, it can be argued that DMSO is the least suitable solvent for further biological studies of nitroxide-containing complexes.

At the same time, based on the EPR spectra of solutions of the Eu(III) complex **10a** in THF, as well as in deionized water and D_2_O (Appendix A), the nitroxide radicals in these solvents do not undergo any changes and exist for a long period of time.

### 2.4. Photophysical Studies

Due to the high lipophilicity of the 2,2′-bipyridine ligands, the obtained complexes **10** have excellent solubility in DCM, and this solvent was selected for the photophysical studies of all the complexes **10**. In UV spectra in all the cases, these complexes exhibited strong absorption maxima in the region of 227 and 311 nm (Figure 4 and Table 1). The longest wavelength maximum is comparable with those for the previously reported europium(III) cation complexes **11a** and **11b** [38]. In addition, complexes **10** demonstrate a bathochromic shift of the absorption maxima compared to the 5-(4-methoxyphenyl)-containing ligand **11c**, from 295 to 311 nm.

Upon the excitation of all the Ln(III) complexes at the absorption maxima only for the Eu(III) complex **10a**, a well-pronounced characteristic lanthanide cation luminescence was observed. As a result, in the luminescence spectrum, all the characteristic emission bands were clearly observed, which correspond to the electronic transitions, namely, ^5^D_4_→^7^F_1–4,_ and all these bands correspond to the emission maxima of Eu(III) cation (590, 617, 652, 695 nm) [53,54] (Figure 5). This confirms the effectiveness of the energy transfer from the TEMPO-bipyridine ligand to the chelated Eu(III) cation. For the complexes of Sm(III) and Gd(III), no luminescence was observed, probably due to the values of the T_1_ energy of bipyridine ligands not falling within the required range. In the case of the Tb(III) complex **10b**, only a very weak luminescence of the terbium(III) cation was observed, which did not allow a reliable measurement of the luminescence quantum yield of this cation. Similar behavior was described in the literature [55,56]. Some similar complexes are presented in Figure 6. All the results of the photophysical studies are combined in Table 1.

The luminescence quantum yield of the Eu(III) cation for the chelate **10a** was found to be about 2.4%, which is somewhat lower than for the previously described complexes **11**. This can be explained by the presence of free radical moieties in the 2,2′-bipyridine ligands. In the literature, only a few examples of the influence of the oxo radical moieties in the composition of the lanthanide complexes on the luminescent properties were reported [52]. Thus, an example of a significant increase in the luminescence intensity of the Tb(III) complex as a result of recombination of the nitroxide fragment of the TEMPO moiety with C-centered radicals in DMSO medium was reported [51]. In the opposite case, during the formation of the nitronyl nitroxyl radical moieties in DOTA-containing ligand in Eu(III) and Yb(III) complexes, a dramatic quenching of the luminescence of these cations, namely by 15 and 28%, was observed [58]. In addition, up to 95% quenching of the luminescence of Tb(III) and Yb(III) lanthanides was observed upon the electrochemical oxidation of their complexes. In this case, the decrease in the emission intensity is associated with the formation of nitroxyl radicals in ligands coordinated with the lanthanide (III) cation [59].

Thus, compound **10a** can be considered a “pre-luminescence” probe suitable for the photophysical studies of samples containing S-containing amino acid residues, such as proteins and oligopeptides.

### 2.5. Fluorescence and EPR Titration with Thio Compounds

Further studies of the photophysical properties of complex **10a** were aimed at studying the interaction between nitroxyl radical moieties and thiol moieties of amino acids. Thus, when the THF solution of compound **10a** (10^−5^ M) was treated with *L*-cysteine (five-fold excess was introduced to react completely with all three TEMPO moieties), a significant increase in the luminescence intensity of the Eu(III) cation was observed (Figure 7a). The same effect was observed in the presence of glutathione (Figure 7b).

To confirm the reduction of the nitroxyl radical in the composition of the complex **10a** in the presence of *L*-cysteine, a number of EPR experiments were carried out. Figure 8 shows a series of EPR spectra of a THF solution of complex **10a** (10^−3^ M) and mixtures thereof with a five-fold excess of *L*-cysteine. Immediately after the addition of *L*-cysteine, significant degradation of the nitroxyl radicals was observed, which was expressed in a drop in the signal amplitude by more than five times. Further registration of the EPR spectra of the reaction mixture illustrates a significant degradation of nitroxyl radicals within 20 min, with the effect ceasing 40 min after the addition of the above-mentioned amino acid. Additionally, the EPR titration of compound **10a** with *L*-cysteine shows a proportional attenuation of the EPR spectra without changing their shape.

The presented data confirm the influence of certain biogenic thiols, such as *L*-cysteine and glutathione, on the luminescence intensity of the Eu(III) complex **10a** in THF solution, as well as on its electron paramagnetic properties, such as the intensity of the free radical signal. The main reason for this may be due to the addition reaction between thiol and free radical moieties, such as TEMPO, with/or without further oxidation [37]. As a result, the photoluminescence-quenching effect of these free nitroxide radical moieties disappears. A similar quenching effect of nitroxide radicals was observed in the case of some common organic fluorophores, such as pyrene [60].

### 2.6. Tests on Tissue Sections of Mice with Amyloidosis

The staining of brain slices of 5xFAD mice (genetic determinate amyloid pathology) line was performed. This line is a model of a rapidly increasing pronounced amyloidosis. A standard dye, Congo red, was used to clarify the amyloids’ position in the cells. Additionally, the nuclear dye DAPI was used to visualize the nuclei and nucleoli. As a result, Congo red stained the cytoplasm and demonstrated the accumulation of the amyloids in the cytoplasm of cells against a background of clearly localized nuclei (Figure 9).

Other samples were stained simultaneously by the nuclear dye DAPI and by the Eu(III) complex **10a**. Under all excitation wavelengths, only DAPI fluorescence was visible in the nuclei. No clusters morphologically similar to amyloid were observed. At the same time, the color of the cytoplasmic membranes of cells was also not fixed (Figure 9).

Due to the lack of a pronounced response to the amyloid in tissue samples and keeping in mind the observed earlier possible interaction with Eu(III) complex **10a** cysteine during fluorescent titration as a next step, we studied the possibility of using Eu(III) complex **10a** for the staining of a tissue of an outbred rat’s liver fixed with cysteine and glutathione.

Usually, tissue slices exhibit a noticeable autofluorescence, which is always greater than that of the living cells. In most cases, this autofluorescence has a low influence on the results of the photophysical studies. However, for studying weakly fluorescent substances, autofluorescence can be an obstacle. That is why, as a next step, we compared the test samples with the control tissue samples, such as the rat’s liver samples.

In Figure 10, fixed histological sections of rat’s liver tissue, stained with the complex **10a** as well as unstained ones under the lambda mode of a confocal microscope, are presented. The images were taken by using the same settings for the laser intensity and other parameters. As a result, a significant increase in the intensity of the tissue fluorescence was observed. However, no selectivity in the staining was detected, and the resulting fluorescence was uniform over all the histological section regions.

### 2.7. Vero Cell Culture Tests

As a next step, Vero culture cells were stained by the complex **10a** without fixation, and, depending on the excitation wavelength, different results were observed. In the picture below, the pictures of cells stained with substance **10a** taken at the different excitation wavelengths are presented (Figure 11). As can be seen, a noticeable autofluorescence of living cells takes place; however, the Eu(III) complex **10a** emission exceeds sufficiently in intensity.

According to the pictures above, the cells stained with **10a** showed the brightest fluorescence upon the excitation with a 488 nm laser, with a maximum fluorescence intensity occurring at 530–540 nm. At the same time, the picture obtained with the excitation at 488 nm is sufficient to obtain an image with a relatively good resolution. However, this resolution is not the best one, probably, due to the insufficient fluorescence extensivity.

To complete the visualization of the staining, the channel mode of the microscope was used, and the resulting fluorescence was accumulated and summarized in the range from 500 to 650 nm. The focus of the microscope was concentrated on the layer of cells adjacent to the substrate. The image was displayed in artificial colors, allowing it to obtain the maximum contrast of small details. Based on the analysis of the obtained image, the Eu(III) complex **10a** accumulates in the cytoplasmic membrane of cells, while the internal staining structures were not observed (Figure 12).

Next, the Hanks’ solution enriched with cysteine was added to the cells stained with the complex **10a**. As a result, no changes are observed (Figure 13).

As the last step, some possible reasons for the differences in the color of the sections with the spectrophotometric data were suggested.

### 2.8. Studies of Interaction between the Complex 10a and BSA by Means of EPR and Photophysical Methods

Based on the photophysical studies, biological staining results and EPR experiments, one may suggest the occurrence of some non-covalent interactions between the Eu(III) complex **10a** and proteins, which results in the changes in the emission of **10a**. To check that the interaction between the Eu(III) complex **10a** and *Bovine Serum Albumin* (BSA), a model carrier protein [61,62,63,64], was studied. Thus, in the aqueous solution of BSA (2 × 10^−6^ M) an intensive fluorescence quenching was detected upon the addition of **10a** in the concentration range 10^−5^–10^−4^ M. In UV-titration experiments at the molar ratios of BSA: complex **10a** as order 1:0 (control sample), 1:5, 1:10, 1:15, 1:20, 1:25, 1:30, and 1:50, an increase of the optical density of the maximum near 275 nm was observed along with the appearance of a new absorption maximum in the range of 300–310 nm, which probably corresponds to the intrinsic absorption of the complex “BSA: **10a**” (Figure 14). At the same time, the optical density of the BSA absorption maximum (278 nm) does not decrease relative to the longer wavelength maximum, which gives grounds to assert that there is no chemical interaction between BSA and the Eu(III) complex **10a**. In addition, based on the data of EPR spectroscopy in aqueous solutions of the complex **10a** and its mixture with BSA, taken in the concentrations equal to those at the endpoint of the titration, no attenuation of the characteristic signals of nitroxyl radicals was observed, which confirms the absence of the chemical reaction between the **10a** and BSA.

According to the emission spectra, the quenching of the intrinsic fluorescence of BSA was observed (emission maxima at 336, 557, and 702 nm) in solution with an increase in the concentration of the complex **10a**. Moreover, the least intense BSA emission maximum (557 nm) demonstrates a bathochromic shift to 578 nm and a 12-fold decrease in intensity upon the increase of the concentration of the complex **10a**. Thus, there is a reason to believe that complex **10a** is bound by BSA by means of intercalation (Figure 15).

The most characteristic criteria for evaluating the effectiveness of the formation of non-fluorescent complex “**10a**: BSA” is the value of the Stern--Volmer constant, quenching constant/association constant, which is expressed by the equation:I_0_/I = 1 + Ksv·[Q],
where I_0_, I are the fluorescence intensity before and after the addition of the Eu(III) complex **10a** (quencher); [Q] is the concentration of the complex compound, mol/l; and Ksv is the quenching constant value, M^−1^.

Essentially, the quenching process can include two components: static quenching (formation of a non-fluorescent complex in the ground state) and dynamic quenching (quenching as a result of the “quencher–fluorophore” collision) [65]. During dynamic quenching, the quencher diffuses to the fluorophore in the excited state, and this fluorophore returns to the ground state without emitting a photon. During the static quenching, a non-fluorescent “quencher: fluorophore” complex is formed [65]. The type of quenching can be estimated by plotting I_0_/I versus [Q]. Based on the graph below (Figure 15B), the experimental and approximated Stern--Volmer plots change according to a parabolic law, and the calculated value of the Stern--Volmer constant was as high as 5.93 × 10^10^ M^−1^ (R^2^ = 0.998), which corresponds to the values observed earlier for BSA, such as 10^4^–10^10^ M^−1^ [66]. This result also indicates a high degree of binding of the Eu(III) complex **10a** to this protein with the predominance of static quenching [64].

The static type of quenching was also confirmed by the changes in the absorption spectra as described above. Thus, with an increase in the concentration of the Eu(III) complex **10a**, noticeable changes in the BSA absorption band (278 nm) were observed along with the appearance of a new band, which indirectly confirms the formation of the complex “**10a**:BSA” in the ground state [64].

In addition, based on the data of the EPR spectroscopy (Figure 16) of aqueous solutions of the Eu(III) complex **10a** and its mixture with BSA, with concentrations equal to those at the endpoint of the titration, no attenuation of the signals characteristic of the presence of nitroxide radicals was recorded, which confirms the absence of the chemical reaction of the latter with BSA.

The results of the study suggested the formation of an inclusion complex between the Eu(III) complex **10a** and BSA. In this case, apparently, the aromatic 2,2′-bipyridine fragment, as well as the hydrophobic part of the ligand, are responsible for the formation of the inclusion complex. Taking into account the well-known BSA binding sites, one can assume the involvement of drug-binding fragments-IIA and IIIA [67]. The binding of an iron(III) complex and a 2,2′-bipyridine ligand to domain II and domain IIIA of BSA [68], as well as a nickel(II) complex to the same domains [58], has already been reported. The literature also describes the influence of the cysteine-copper complex on the shape of the final agglomerate but with the preservation of the structure of the BSA itself [69].

## 3. Materials and Methods

### 3.1. Chemicals and Instruments

UV-vis absorption spectra were recorded on the Shimadzu UV-2550 spectrophotometer. Emission and excitation spectra were recorded on the Horiba FluoroMax-4 spectrofluorometer. Absolute quantum yields were obtained using the Integrating Sphere Quanta-φ of the Horiba-Fluoromax-4. Time-resolved fluorescence measurements were carried out using time-correlated single-photon counting (TCSPC) with a nanosecond LED (370 nm). 1HNMR and 13CNMR spectra were recorded on the Bruker Avance-400 spectrometer at 298 K using tetramethylsilane (TMS) as an internal standard. Mass spectra were recorded on GCMS-QP2010 Ultra (Shimadzu) and Bruker maXis Impact HD for the HR-mass measurements, and electrospray was used as a method of ionization. Microanalyses (C, H, N) were performed using a Perkin–Elmer 2400 elemental analyzer. TLC was performed on a silica gel-coated aluminum slide (Merck, Silica gel G for TLC). Silica gel (60–120 mesh, SRL, India) was used for column chromatography. Melting points were measured on the instrument Boetius. All solvents were dried and distilled before use. Solvents, reagents and chemicals were purchased from Aldrich, Fluka, Merck, SRL, Spectrochem and Process Chemicals. All reactions with moisture-sensitive reagents were carried out in oven-dried glassware.

EPR spectra were measured on a Bruker Elexsys E500 spectrometer using a TMTH spin probe (N-(1-Hydroxy-2,2,6,6-tetramethyl piperidine-4-yl)-2-methylpropanamide). Substances were dissolved in deionized water at a concertation of 2 mM.

TLC was performed on a silica gel-coated aluminum slide (Merck, Silica gel G for TLC). Silica gel (60–120 mesh, SRL, India) was used for column chromatography.

All solvents were dried and distilled before use. Commercially available substrates were freshly distilled before the reaction. Solvents, reagents, and chemicals were purchased from Aldrich, Fluka, Merck, SRL, Spectrochem and Process Chemicals. All reactions involving moisture-sensitive reactants were executed using oven-dried glassware.

### 3.2. Biological Studies

#### 3.2.1. Ethics

The study has been conducted on animals that were lawfully acquired. The experimental procedures involving animals were in compliance with the applicable laws and regulations as well as the principles expressed in the National Institutes of Health, USPHS and Guide for the Care and Use of Laboratory Animals.

Non-transgenic Female Wistar rats (16-weeks old) were obtained from the Institute of Immunology and Physiology, the Ural Brunch of RAS (Yekaterinburg, Russian Federation). The animals were kept under equal conditions (12 h light/12 h dark cycle with lights turned on at 9:00 a.m.; temperature 20 ± 2C), were housed 5 animals per cage and were fed according to the customary schedule with free access to water. The animals showed no symptoms of any disease.

A median laparotomy was performed under general anesthesia. The stomach, the spleen and structures of the pancreas adjacent to the stomach in the projection of large curvature along with the fatty tissue were extracted. The samples of pancreatic tissue were separated from the fatty tissue and immersed in 10% neutral formalin for 24 h at room temperature. The fixing solution was then replaced by paraffin through a series of solutions that included solutions of alcohol in increasing concentration (50%, 70%, 95% of absolute ethanol (3 sequences), followed by 3 sequences of xylol and 2 sequences of hot paraffin. The preparation of samples for histological examination was carried out using an automatic processor Leica EG 1160, followed by paraffin-embedding. After microtomy, the sections were placed in a container with water and then on slides coated with an adhesive composition. The paraffin was removed from serial sections 3–4 mm thick. Slides were placed sequentially in xylol, 100% ethanol and in solutions with a gradual decrease in the concentration of alcohol to a completely aqueous solution. Recruitment of macrophages and bone marrow stem cells were to regenerate. Next, without drying off the slides, each serial section was stained with ethanol 3.8 mmol/L or DMSO solution 6.8 mmol/L of substance.

Transgenic mice were obtained from the Center for Collective Use of IPAC RAS. Animals were housed in groups of five per cage in a standard environment (12-h light/dark cycle, 18–26 °C room temperatures and 30–70% relative humidity) with food and water ad libitum. The procedures were carried out in accordance with the “Guidelines for accommodation and care of animals. Species-specific provisions for laboratory rodents and rabbits” (GOST 33216-2014) and were in compliance with the principles enunciated in the Directive 2010/63/EU the protection of animals used for scientific purposes and were approved by the local Institute of Physiologically Active Compounds Ethics Review Committee (protocol #52, 18 September 2020). Prior to collecting tissues, animals were terminally euthanized, followed by a brain necropsy. For the histological study, mice brains were fixed in 10% neutral buffered formalin (Leica Biosystems Inc., Deer Park, USA) at +4 °C overnight. Mice brain dehydration in ethanol-xylene series was according to the following scheme: deionized water (2 h); 70% ethanol (12 h at +4 °C); ethanol 96% sequentially (5 min, 15 min, 4 stages × 10 min each); a mixture of ethanol and xylene 1:1 (30 min); xylene (2 stages × 30 min each, fresh xylene overnight at +4 °C); and paraffin (3 stages × 1 h each) in a Leica ASP200 apparatus (Leica Biosystems Inc., Deer Park, USA). See details in [70].

#### 3.2.2. Cell Cultivation

The ability of substances to stain cells for fluorescence microscopy was also investigated. We used a Vero cell line obtained from the cell collection of Biolot (Saint Petersburg, Russia). Cell culture is maintained in culture flasks (Eppendorf, Vienna, Austria), in DMEM (Sigma-Aldrich, St. Louis, USA) supplemented with 10% fetal calf serum (Biolot, Saint Petersburg, Russia) and 0.5% gentamicin (Biolot, Saint Petersburg, Russia) in an incubator with atmosphere 5% CO_2_.

#### 3.2.3. Cell and Tissue Staining

For staining by substances, the cells were diluted to a concentration of 10^4^ cells per mL and transferred to glass-bottom dishes (Jet Biofil., Guangzhou, China), where they were cultivated for 24 h. Then the nutrient medium was changed, and the substance solution in DMSO (at a concentration of 25 mmol/L) was added in an amount of 50 μL per 1 mL of the nutrient medium. Standardly, the cells were incubated with the test substances for 30 min. However, an experiment was also conducted where the substance was added to the cells directly in the process of microscopic observation. There was no significant difference in the fluorescence spectra or in the distribution of the substance over the cells depending on the incubation time.

To stain the histological sections of the brain, a dye solution was added to the deparaffinized section, after which the section was kept with the dye for 30 min. Then the sample was washed in distilled water for 10 min, followed by a DAPI solution added at a concentration of 200 nM. Next, the sample was washed for 2 min in distilled water, after which it was dehydrated in alcohols, cleared in xylene and placed in a transparent medium for observation.

#### 3.2.4. Microscopic Examination

Microscopic examination was performed using the equipment of the Shared Research Center of Scientific Equipment SRC IIP UrB RAS. After staining and washing, living cells were examined using a confocal laser scanning microscope LSM-710; Carl Zeiss has a multichannel QUASAR detector (34 channels). The images were obtained using an immersion lens 40x/1.3 Oil. To obtain an informative fluorescent image in special software ZEN, a special lambda mode (λ-mode) was used, which allows for determining the emission range with the maximum contrast for this preparation. The studies were carried out upon excitation by a laser with a wavelength of 405 nm, and the emission was recorded in the entire range of the confocal microscope (400–750 nm). The emission spectra of substances were also extracted from images obtained in the lambda mode. However, it is necessary to clarify that the confocal microscope is not a spectrofluorometer, and the fluorescence spectra obtained with it can be unreliable.

Histological sections of the pancreas and brain were also examined in lambda mode.

The relative fluorescence intensity of the sample was determined from the maximum fluorescence intensity extracted from the image obtained in the lambda mode. For this, the intensity spectrum was determined at ten points of the image, the maximum value was averaged, and confidence intervals were calculated. Since the images were taken at different settings, the value was adjusted for the laser power and the height of the confocal cut.

The images were processed using LSM Image Browser, ImageJ and a custom Python script that uses a napari library to work with image data [70].

## 4. Conclusions

New Eu(III), Tb(III), Gd(III) and Sm(III) complexes based on 5-phenyl-2,2′-bipyridine-6-carboxylic acid containing TEMPO residues have been obtained. Based on the presence of a characteristic lanthanide luminescence, the Eu(III) complex **10a** was selected as a probe for the photophysical studies and biological staining studies. It was found that, upon the interaction of this Eu(III) complex with free biogenic thiols, such as cysteine and glutathione, a characteristic luminescence of the Eu(III) cation increases. It was suggested that this luminescence increase is a result of the improved energy transfer from the 2,2′-bipyridine ligand to the Eu(III) cation due to the recombination of nitroxyl radical fragments in the TEMPO moieties. The disappearance of the signals of nitroxyl radicals of TEMPO was confirmed by the data of EPR experiments for complex **10a** in the presence of cysteine and glutathione. However, in the experiments on staining brain sections of mice with severe amyloidosis with the above-mentioned complex, no changes in the photophysical signals were detected. A weak fluorescence was observed upon staining rat liver tissues with the Eu(III) complex **10a**, while no fluorescence response was observed in the case of either *Vero* cells or those in the cysteine-enriched medium. Based on the results of fluorescence titration of BSA, a model protein with the Eu(III) complex **10a**, a static quenching of the (auto)fluorescence of the BSA was revealed. The formation of a non-covalent non-fluorescent inclusion complex “**10a**: BSA” was suggested, and its Stern-Volmer static quenching constant as high as 5.93 × 10^10^ M^−1^ was calculated.

## Data Availability

Not applicable.

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
