# Peer review of "New TEMPO–Appended 2,2′-Bipyridine-Based Eu(III), Tb(III), Gd(III) and Sm(III) Complexes: Synthesis, Photophysical Studies and Testing Photoluminescence-Based Bioimaging Abilities"

_molecules, 2022, doi:10.3390/molecules27238414_

Round 1

Reviewer 1 Report

The paper of Emiliya V. Nosova and co-authors is a fundamental work on synthesis new Ln(III) compounds with free radicals and studying Eu(III) derivative bioimaging abilities. The authors made a great job, organic presynthesized compounds were characterized with NMR and EPR, copper compound (9) with EPR and ESI-MS, Ln(III) compounds with EPR, ESI-MS and elemental analysis.

The yields of the reaction products are quite high (10 a-d), but the amount is 14-27 milligrams. Is this amount sufficient for the experiments carried out in the article? If not, how is the isostructurality and purity of the resynthesized compounds confirmed?

The composition of the newly obtained compounds is confirmed by mass spectra, elemental analysis and the presence of radicals is confirmed by the EPR method. And how is the purity of samples confirmed for impurities?

The procedure for the synthesis of substances 10 a-d indicates that the compounds are extracted with methylene chloride. If the substances are soluble in organic solvents, why weren't single crystals suitable for X-ray diffraction analysis isolated? Then it would be possible to make X-ray phase analysis for the presence of impurities.

Is it possible to make x-ray phase analysis for compounds 10 a-d to confirm their isostructurality?

Line 195: PR

Line 216: carried out?

Line 247: he excitation

Line 502: Celsius degree sign

Line 575-576: Describe what is contained in the supplementary materials. Also add information to the manuscript that the experimental section for the synthesis of target compounds is contained in additional materials. And add the product reaction yields to Scheme 1.

The work is of great interest for the chemistry of lanthanides and free radicals. I believe that the work will be well cited and recommend it be accepted into Molecules.

Author Response

The yields of the reaction products are quite high (10 a-d), but the amount is 14-27 milligrams. Is this amount sufficient for the experiments carried out in the article? If not, how is the isostructurality and purity of the resynthesized compounds confirmed?

Comments: The synthesis of compounds 10a-d was repeated 3 times to collect the amount of the substrates for carrying out all the described experiments. The isostructurality and purity of the resynthesized compounds were confirmed each time by means of 1H NMR spectroscopy and the elemental analysis (please, see ESI for details).

The composition of the newly obtained compounds is confirmed by mass spectra, elemental analysis and the presence of radicals is confirmed by the EPR method. And how is the purity of samples confirmed for impurities?

Comments: The purity of samples for impurities is confirmed by means of elemental analysis.

The procedure for the synthesis of substances 10 a-d indicates that the compounds are extracted with methylene chloride. If the substances are soluble in organic solvents, why weren't single crystals suitable for X-ray diffraction analysis isolated? Then it would be possible to make X-ray phase analysis for the presence of impurities.

Comments: Only very small crystals of compounds 10a-d were obtained after the slow evaporation of samples from dichloromethane and other solvents. Unfortunately, the size and the shape of the crystals were not suitable for carrying out X-ray diffraction analysis.

Is it possible to make x-ray phase analysis for compounds 10 a-d to confirm their isostructurality?

Comments: The answer was given above.

Line 195: PR

Comments: Corrected EPR

Line 216: carried out?

Comments: Corrected

Line 247: he excitation

Comments: Corrected "the"

Line 502: Celsius degree sign

Commnets: Corrected

Line 575-576: Describe what is contained in the supplementary materials. Also add information to the manuscript that the experimental section for the synthesis of target compounds is contained in additional materials. And add the product reaction yields to Scheme 1.

Comments: Corrected.

Reviewer 2 Report

Review is attached as a PDF file.

Author Response

  1. I suggest to rephrase the sentence Therefore, these solvents may be recommended as the most suitable for the biological studies. The point is that water is a natural environment for biological studies, but not THF or D2O. Anyhow, the stability of of Eu complexes is nice.

Comments: The sentence was deleted.

  1. In the biological tests of dyes the time was not given. Since the stability of a dye is important it is worth to mention after what time of incubation the pictures shown in Fig. 9 were taken.

Comments: In a typical experiment the cells were incubated with the test substances for 30 minutes. However, an experiment was also conducted where the substance was added to the cells directly in the process of microscopic observation. There was no significant difference in the fluorescence spectra or in the distribution of the substance over the cells depending on the incubation time. This text has been added to the article materials and methods.

  1. It should be explained what the numbers at Fig. 11 are. Are those the emission or excitation wavelengths? It would be strange if those are excitations as compound 10a does not absorb below 400nm (Fig. 4).

Comments: The wavelengths are excitation ones. In biological conditions an exciplex emission or another effects could be observed. That’s why we always check images on all wavelengths.

  1. I can not see any results related to pure fluorophoe-β-amyloid interaction. The use of cells from animals does not warrant any results come from specific interaction and the conclusions may be wrong. Thus, I suggest to weaken the introduction pointing towards the AD as the main goal here.

Comments: It must be admitted that in the end our approach to the detection of amyloid in biological tissues did not work, probably due to the instability of the substance, which was shown by the results of histological experiments. However, we believe that the publication of negative results is just as important, since the publication of only positive results negatively affects modern science. As it was suggested, we have weaken the introduction and made it more broad.

  1. In the light of interaction of sulfur atoms present in proteins I suggest to add the following reference: Coordination Chemistry Reviews 367 (2018), 18-64.

Comments: This reference was added.

  1. In Fig. 5 one of bands observed may come form doubled the excitation wavelength. That, for sure, needs explanation.

Comments: The emission spectrum of compound 10a contains all bands corresponding to electronic transitions of the Eu(III) cation. It is not entirely clear to the authors which of the observed bands is involved, given that the excitation wavelength is 305 nm.

  1. The bands present in Fig. 15 close to 550-575 nm look like the scattering of the excitation light but not the emission. I’d be happy if authors comment on that.

Comments: During fluorescent titration, the intensity of scattering bands in fluorescence spectra, as a rule, does not decrease with a change in the concentration of compounds. In our case, (Fig. 15) in the range of 550-575 nm, a decrease in the intensity of the maxima with an increase in the concentration of the complex in the solution is observed, similar to the main maxima (336 and 702 nm). Thus it can be argued that, in this case the presence of additional emission bands, and not scattering on this interval.

  1. The manuscript does not describe the stability of obtained complexes in detail. Is that possible

to study those based on NMR techniques?

Comments: Unfortunately, all the compounds 8-10 containing the nitroxyl radicals are paramagnetic and cannot be characterized by 1H NMR.

  1. The link present in reference 26 is wrong. It leads to graphics. (10.1021/OL035589W/ASSET/IMAGES/MEDIUM/OL035589WN00001.GIF)

Comments: The Link was corrected.

  1. The additional NMR spectra need to be recorded. That is, for example, 13C spectrum for 7. In Fig. S3 it is clearly see that the concentration is very high taking into account the residual signal form chloroform at ca. 7.26ppm. In my opinion for remaining compounds it is still possible when one look at the 1H spectra.

Comments: Additional 13С NMR spectra for compounds 6 and 7 are given in supporting information. The values of CDCl3 signals are due to the relative solubility of the compounds in them.